# (In-Vitro Comparison between Closed Versus Open CAD/CAM Systems) Comparison between Closed and Open CAD/CAM Systems by Evaluating the Marginal Fit of Zirconia-Reinforced Lithium Silicate Ceramic Crowns

**Gil Ben-Izhack** *,† ⓘ, **Asaf Shely** † ⓘ, **Omer Koton**, **Avi Meirowitz**, **Shifra Levartovsky** ⓘ and **Eran Dolev** ⓘ

Department of Oral Rehabilshitation, Goldschleger School of Dental Medicine, The Sackler Faculty of Medicine, Tel Aviv University, Tel Aviv 6997801, Israel; asafshely@gmail.com (A.S.); dr.omerkoton@gmail.com (O.K.); Eintmz@post.tau.ac.il (A.M.); shifralevartov@gmail.com (S.L.); eran@drdolev.com (E.D.)

\* Correspondence: gil.ben.izhack@gmail.com; Tel.: +97-254-598-6343

† These authors contributed equally to the work.

**Abstract:** Background: This study compared the marginal gap (MG) and absolute marginal discrepancy (AMD) of computer-aided design and computer-aided manufacturing (CAD–CAM) used in open systems (OSs) and closed systems (CSs) for producing monolithic zirconia-reinforced lithium silicate (ZLS) ceramic crowns. Methods: 60 ZLS ceramic crowns were cemented to abutment acrylic teeth; thirty crowns were designed and milled by an OS, and thirty by a CS. All crowns were sectioned for evaluating the marginal gap by scanning electronic microscopy (SEM). To compare the marginal gap between CS and OS techniques, data were analyzed using the independent-samples Mann–Whitney U Test ($\alpha = 0.05$). Results: AMD was found to be significantly better for the closed system ($p < 0.05$). Mean AMD values for the CS were 148 μm, and for the OS it was 196 μm. MG was found to be significantly better for the OS ($p < 0.05$). Mean MG values for the CS were 55 μm, and for the OS they were 38 μm. Conclusions: The marginal gap in relation to AMD was significantly better for CS. However, the marginal gap in relation to MG was significantly better for OS. Both techniques showed clinically acceptable MG values (<120 μm).

**Keywords:** CAD–CAM; ZLS; marginal fit; CEREC; marginal gap; open system; closed system

## 1. Introduction

Today, there is an increasing demand for metal-free restorations. The use of all ceramic restorations is well accepted because of improved esthetic and mechanical characteristics [1,2]. ZLS is a metal-free material which can be used due to the progression of CAD–CAM technology [3].

CELTRA® DUO (Sirona Dentsply, Milford, DE, USA) is a CAD–CAM material which is classified as a zirconia-reinforced lithium silicate. The matrix of the material is silica-based glass (58%) with dissolved particles of zirconia (10%). The manufacturer recommends sintering for greater flexural strength, although it is not mandatory [3]. There are two phases of CELTRA® DUO microstructure; one is lithium metasilicates ($Li_2SO_3$), the other is lithium orthophosphates ($Li_3PO_4$) [4].

It is well known that the success and longevity of fixed partial dentures (FPDs) are influenced by several factors, including the marginal gap. Microleakage is the main reason for failure of FPDs because it may lead to secondary dental caries, endodontic lesions, periodontal disease, and bone loss [5,6]. In the literature, a marginal gap up to 120 μm is acceptable; some studies have proposed one even less, of 100 μm, for better survival of the FPD [7,8]. A study by McLean which investigated marginal discrepancy in more than 1000 crowns showed that increased longevity is associated with marginal discrepancy of less than 120 μm [9].

CAD–CAM can be used by two different types of systems. The first is a closed system in which the dentist scans intra-orally, designs, and mills the restoration chair-side with one company flow chart. The second is an open system in which the dentist can scan with any intra-oral scanner, then send the STL file to a laboratory where it is designed and milled [10,11]. Studies on the marginal fit of closed systems have reported acceptable marginal discrepancies of 81 μm and 87 μm [12,13]. For open systems, studies have shown marginal discrepancies between 51 μm and 90 μm [14,15].

In the literature, there are many methods which describe how to measure marginal fit. Holmes defined the marginal gap (MG) as a vertical line between the restoration to the preparation margin. AMD is measured horizontally from the margin of the restoration to the preparation. The risk of microleakage increases as the MG is increased. AMD represents under- or over-extension of the restoration, which can lead to an increased risk of plaque accumulation [16].

For measuring marginal gaps, there are several methods such as micro-CT, sectional technique, and silicone paste [8,17–20]. For evaluating the marginal gap (MG and AMD) we used the sectional method.

To the best of our knowledge, there are no studies which have compared the effects of closed systems versus open systems on the marginal discrepancy of ZLS crowns. The purpose of this in vitro study was to compare OS versus CS by evaluating the marginal discrepancy of ZLS crowns. The null hypothesis was that no difference would be found between OS and CS techniques.

## 2. Materials and Methods

The sectional method has been previously described by Dolev et al. [12,21] for evaluating the marginal gap: 60 typodont maxillary right first molars (FLUX 8634; Columbia Dentoform, Lancaster, PA, USA) were used. The CS group included 30 typodont teeth scanned with an intraoral scanner (CEREC® AC Omnicam; Dentsply Sirona, Milford, DE, USA), the finish line was marked, and the restoration was designed (CEREC® SW 4.5.2; Dentsply Sirona, Milford, DE, USA) and milled from zirconia-reinforced lithium silicate ceramic blocks (CELTRA® DUO, Sirona Dentsply, Milford, DE, USA) using a CAM milling unit (CEREC inLab MC XL®; Dentsply Sirona, Milford, DE, USA) (Figure 1).

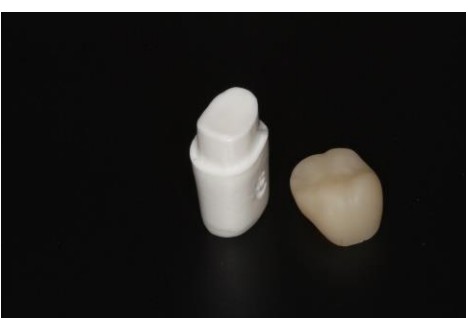

**Figure 1.** First maxillary right molar tooth and zirconia-reinforced lithium silicate crown.

The open system group included 30 typodont teeth which were scanned with an intraoral scanner (CEREC® AC Omnicam; Dentsply Sirona, Milford, DE, USA). The 30 scans were converted from RST files, which are specific for CEREC® software, to universal STL files, using a CEREC® Dongle, and data were imported to an open CAD system (EXOCAD® DentalCAD 2.2 Vallenta, Darmstadt, Germany). The finish line was marked, and the restoration was designed. Crowns were milled from zirconia-reinforced lithium silicate ceramic blocks, CELTRA® DUO (Sirona Dentsply, Milford, DE, USA), using a CAM milling unit (CEREC inLab MC XL®; Dentsply Sirona, Milford, DE, USA).

Specific parameters for all 60 crowns included in this study were identical, as follows: radial spacer—120 μm; occlusal spacer—120 μm; proximal contacts—25 μm; radial minimal

thickness—800 μm; occlusal minimal thickness—1000 μm; absolute marginal thickness—50 μm; marginal ramp angle—45°; marginal ramp width—150 μm.

All crowns were cemented with self-adhesive resin cement (Rely X U-200; 3M ESPE), while using a pressing machine Lutron Electronic Enterprise Co. Ltd FG-20KG (Taipei City, Taiwan) which exerted a constant force of 50 Newtons on the model and crown at the time of cementation, in order to avoid the effect of diverse compressive forces on the marginal compatibility. At this point, 60 samples (30 for each group) were made up of a custom crown for the model being scanned and pasted into one unit. All units (crown and abutment) were cut by a precision saw (Isomet Plus cutting saw; Buehler), first from buccal to lingual then from mesial to distal, creating four samples from each unit (Figure 2):

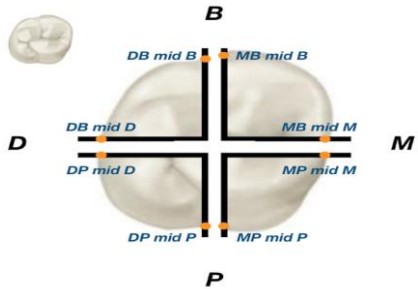

**Figure 2.** A unit with 8 locations of measurements.

mesio-lingual (ML), disto-lingual (DL), disto-buccal (DB), and mesio-buccal (MB). All specimens were evaluated by using a scanning electron microscope (JSM-IT100, JEOL, Peabody, MA, USA) at 250× magnification in two regions of interest. Two parameters were measured in this study: first, in the vertical dimension (MG), which represents the micro gap between the restoration and the preparation; secondly, in the horizontal dimension (AMD), which represents the under- or over-extension of the restoration in relation to the finish line of the preparation (Figure 3).

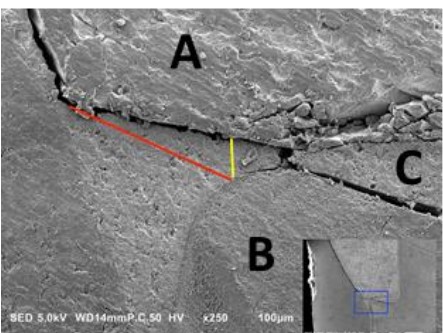

**Figure 3.** A cross-sectional photograph under microscope. Yellow arrow—marginal gap (MG) measurement area. Red arrow—absolute marginal discrepancy (AMD) measurement area. (**A**) Restoration; (**B**) Preparation; (**C**) Cement.

Statistical analysis was performed using the Statistical Package for Social Sciences for Windows Release 23.0 (SPSS Inc., Chicago, IL, USA).

Independent-samples Mann–Whitney U tests were used for the comparison of fit between the production methods. A reliability test was performed by a BH-correct statistical test. The statistical significance level for this work was $p < 0.05$.

## 3. Results

All units were measured in eight locations for the OS group and CS group; mean values and standard errors were calculated for MG and AMD (Figures 4 and 5).

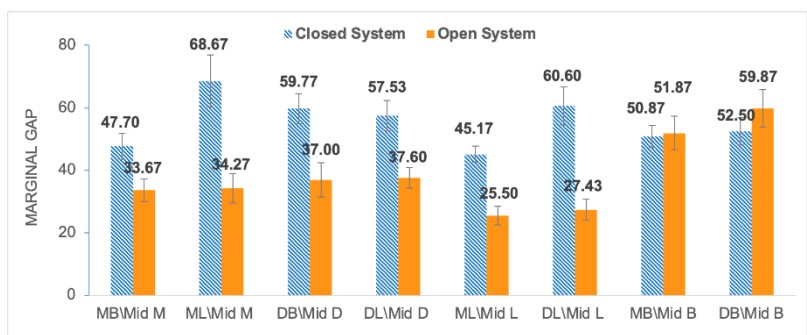

**Figure 4.** Mean values of marginal gap (MG) at 8 locations for the closed system (blue) and open system (orange).

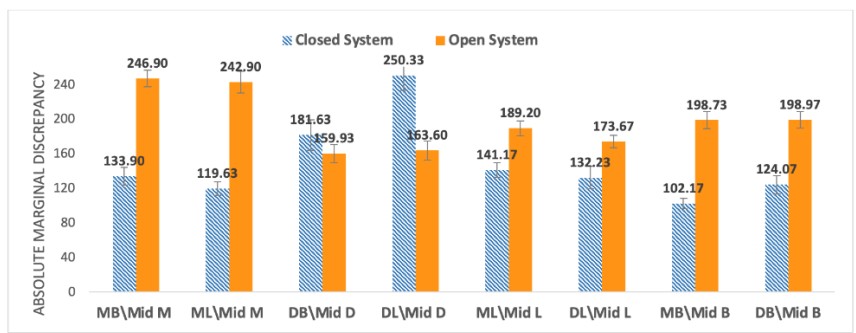

**Figure 5.** Mean values of marginal discrepancy (AMD) at 8 locations for the closed system (blue) and open system (orange).

The open system showed lower MG values compared to the closed system, except for at the MB\Mid B location (Figure 4). The open system showed greater AMD values compared to the closed system, except for in two regions, DL\Mid D and DL\Mid L (Figure 5).

The overall mean ± standard error (SE) value for MG and AMD of the closed system and open system fabrication methods are presented in Tables 1 and 2. We observed statistically significant differences for MG (Mann–Whitney U test; $p < 0.001$; N1 = N2 = 240). We also found statistically significant differences for six out of eight AMD measurements (Mann–Whitney U test; $p < 0.001$; N1 = N2 = 240). The MG 95% confidence intervals are presented in Tables 1 and 2.

**Table 1.** Mean and SE for marginal gap ($\alpha$ = 0.05).

| Method | Mean | SE | 95% Confidence Interval | |
|---|---|---|---|---|
| | | | Lower Limit | Upper Limit |
| Closed system | 55.35 (μm) | 4.839 (μm) | 2.65 (μm) | 8.278 (μm) |
| Open system | 38.4 (μm) | 4.354 (μm) | 2.991 (μm) | 5.913 (μm) |

**Table 2.** Mean and SE for absolute marginal discrepancy ($\alpha$ = 0.05).

| Method | Mean | SE | 95% Confidence Interval | |
|---|---|---|---|---|
| | | | Lower Limit | Upper Limit |
| Closed system | 148.14 (μm) | 11.28 (μm) | 6.129 (μm) | 17.571 (μm) |
| Open system | 196.74 (μm) | 9.84 (μm) | 7.168 (μm) | 12.449 (μm) |

## 4. Discussion

Our null hypothesis regarding MG parameter was rejected; the open system displayed a significantly lower gap ($38.4 \pm 4.4$ μm) compared to the closed system ($55.35 \pm 4.8$ μm). The null hypothesis was also rejected for the AMD parameter, because the open system resulted in a significantly larger gap ($196.74 \pm 9.8$ μm) compared to the closed system ($148.14 \pm 11.3$ μm). It can be clearly stated that there was statistically significant difference ($p < 0.001$) in the marginal fit between methods for both the MG and the AMD measurements.

Different aspects of innovative CAD–CAM systems have been studied. The study group of Beuer et al., who examined the marginal gap in three-unit zirconia frameworks, found lower MG values (29 μm) in the OS (milling center) compared to the values (56 μm) of the CS [19]. Few studies have compared the values of marginal gap between different CS and showed better MG values with the use of CEREC inLAB: Rajan compared two CSs, CEREC versus CERAMILL, and found lower MG for the CEREC [22]; Saab et al. compared four different CSs, CERAMILL, CERCON, LAVA and CEREC and the lowest MG (37 μm) was obtained by CEREC inLAB [23]. Another study by ArRejaie et al. compared three CSs: KaVo, DeguDent and Lava. The lowest MG was achieved by the Lava system (112 μm) [24], although compared to the other studies, this value is higher. Comparisons between the CS and OS are described by Beuer et al., who showed that CSs (Cercon Brain, DeguDent) and OSs (Compartis Integrated Systems, DeguDent) had no significant differences regarding MG [25]. In 2019, Dolev et al. published two studies: one study which compared two techniques (CAD–CAM versus hot-press) for producing monolithic lithium disilicate crowns showed no significant differences between the techniques in relation to the MG [12], whereas in the second, they found that zirconia crowns manufactured with CS (CEREC inLAB) displayed a significantly lower gap ($85 \pm 2$ μm) compared to the OS (LAVA milling center $133 \pm 4$ μm) [21]. However, data on comparisons between closed and open CAD–CAM systems are scarce. In 2018, Kricheldorf et al. published an article examining the MG gaps between a Dentsply Sirona (CEREC) closed system and various open systems [26]. They reported lower mean marginal gaps (MGs) in open systems compared with closed system, thus supporting the results obtained in our study. Nevertheless, there were few limitations to the study, because they used different scanner types for each group, examined only 10 samples for each group, and measured only the vertical marginal interval (MG). However, the advantage in their study was the use of the four-axis milling unit in the closed system and the five-axis milling unit in the open system. As we mentioned earlier, research shows that the five-axis milling unit is significantly more accurate than the four-axis milling unit, and therefore the results of their study are not surprising [18,27].

A systematic review by Abduo et al. [28] presented many methodologies for measuring the marginal fit of monolithic zirconia, which included the section technique [17,29], microcomputed tomography [17,18], and silicone technique [8]. Moreover, parameters such as MG and AMD are described differently between studies, which makes standardization difficult.

Following the conclusions from Kricheldorf's and Abduo's studies, we aimed to control for as many variables as possible in our study: the same researcher performed all model scans, marked the finish lines, made crown designs, and measured all the marginal gaps; we used the same scanner to perform all the scans, the same four-axis milling unit to produce all the crowns, and the same monolithic zirconia-reinforced lithium silicate ceramic block; the crown cementing protocol was the same in each sample and did not rely on a "finger press" as in most previous studies, but on cementing under a pressure device. In addition, we used marginal terminology recommended by Holmes et al. [16] to select the two most relevant areas to determine the marginal compatibility accuracy for each of the groups, marginal gap (MG), and absolute marginal discrepancy (AMD).

Earlier studies showed that the process of cementation increased the marginal gap comparing to the pre-cementation gap [18,30]. In addition, the type of cement also affected the marginal fit according to its components. To overcome those negative effects, we used

the same cement and a special device which applied steady pressure on each sample during cementation, without human intervention.

In this study, the measured MG values (OS 38.4 μm and CS 55.35 μm $p < 0.05$) in both groups were within the clinically accepted values (120 μm) as described by McLean and Fraunhofer [9].

It is important to note that there are no studies which have measured the impact of the conversion of files on their quality. We assume that we can only impair the quality of the digital impression and not improve it during conversion, and therefore the smaller gap observed in the open system is probably related to the improved design capabilities of the open system software. Further research is needed in this area.

## 5. Conclusions

Our study suggests that:

1. There are significant differences in the MG parameter in favor of ZLS crowns produced by open systems compared to closed systems;
2. There are significant differences in the AMD parameter in favor of ZLS crowns produced by closed systems compared to open systems;
3. ZLS crowns fabricated by both closed and open systems showed MG values within the accepted clinically range (120 μm);
4. Our findings suggest that clinicians may choose their preferred manufacturer for each component of the system, without compromising the quality of restorations as a result of data conversion.

**Author Contributions:** Conceptualization, G.B.-I. and E.D.; methodology, A.M.; software, O.K.; validation, G.B.-I., A.S. and E.D.; formal analysis, O.K.; investigation, S.L.; resources, S.L.; data curation, A.M.; writing—original draft preparation, A.S.; writing—review and editing, G.B.-I.; visualization, O.K.; supervision, A.M.; project administration, A.M.; funding acquisition, E.D. All authors have read and agreed to the published version of the manuscript.

**Funding:** This research received no external funding.

**Institutional Review Board Statement:** Not applicable.

**Informed Consent Statement:** Not applicable.

**Data Availability Statement:** Data available on request due to restrictions of privacy. The data presented in this study are available on request from the corresponding author. The data are not publicly available due to privacy.

**Conflicts of Interest:** The authors declare no conflict of interest.

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
