# Peer review of "(In-Vitro Comparison between Closed Versus Open CAD/CAM Systems) Comparison between Closed and Open CAD/CAM Systems by Evaluating the Marginal Fit of Zirconia-Reinforced Lithium Silicate Ceramic Crowns"

_applsci, doi:10.3390/app11104534_

Round 1

Reviewer 1 Report

The research “Comparison between closed versus open CAD/CAM systems by evaluating the marginal fit of zirconia reinforced lithium silicate ceramic crowns” was written according to the instructions.

The introduction gives an insight into the problem. Materials and methods are well described and refer to previous researches. The results are presented with figures and tables. It is recommended to use abbreviations correctly and write the references to the instructions.

Reviewer 2 Report

Dear authors, the present study is well conducted and well written, however i have some comments regarding the manuscript:

-describe more carefully the difference between MG and AMG and the clinical importance of these, focusing on the different way of measuring and reporting these parameters between already published studies (as mentioned in the discussion);

-Improve Figure 3 to clarify what the reader is seeing;

-I suggest to move from the Discussion to Materials and Methods the following part: Page 6 line 188 to 198.

-Highlights in the Conclusion the clinical relevance of your findings;

Best regards.
